# Paclitaxel as HIPEC-Drug after Surgical Cytoreduction for Ovarian Peritoneal Metastases: A Randomized Phase III Clinical Trial (HIPECOVA)

Pedro Villarejo Campos [1,2,*,†], Susana Sánchez García [3,†], Mariano Amo-Salas [4], Esther García Santos [3], Carlos López de la Manzanara [5], Ana Alberca [6], David Padilla-Valverde [3], Francisco Javier Redondo Calvo [7] and Jesús Martín [3]

1. Department of Surgery, Fundación Jiménez Díaz University Hospital, Avda. Reyes Católicos, 2, 28040 Madrid, Spain
2. Department of Surgery, Universidad Autónoma de Madrid, C/Arzobispo Morcillo s/n, 28034 Madrid, Spain
3. Department of Surgery, General University Hospital of Ciudad Real, C/Obispo Rafael Torija, s/n, 13005 Ciudad Real, Spain; susaga@sescam.jccm.es (S.S.G.); epgarcias@sescam.jccm.es (E.G.S.); davidp@sescam.jccm.es (D.P.-V.); jesusmartin57@wanadoo.es (J.M.)
4. Department of Mathematics, University of Castilla-La Mancha, Camino de Moledores, s/n, 13071 Ciudad Real, Spain; mariano.amo@uclm.es
5. Department of Gynaecology, General University Hospital of Ciudad Real, C/Obispo Rafael Torija, s/n, 13005 Ciudad Real, Spain; carlosandres.lcano@uclm.es
6. Department of Surgery, General University Hospital of Jaén, 23007 Jaén, Spain; ana.alberca.sspa@juntadeandalucia.es
7. Department of Anaesthesia, General University Hospital of Ciudad Real, C/Obispo Rafael Torija, s/n, 13005 Ciudad Real, Spain; fjredondo@sescam.jccm.es
* Correspondence: pedro.villarejo@quironsalud.es
† These authors contributed equally to this work.

**Abstract:** Multidisciplinary strategies have transformed the management of advanced ovarian cancer. We aimed to evaluate the effectiveness of paclitaxel in hyperthermic intraperitoneal chemotherapy (HIPEC) following surgical cytoreduction for ovarian peritoneal metastases in a randomized phase III trial conducted between August 2012 and December 2019. Seventy-six patients were randomized to either the HIPEC or no HIPEC group. Although median values for the primary endpoints (recurrence-free survival (RFS) and overall survival (OS)) revealed superior outcomes for the HIPEC (RFS: 23 months, OS: 48 months) over the control group (RFS: 19 months, OS: 46 months), these differences were not statistically significant ($p = 0.22$ and $p = 0.579$). Notably, the HIPEC group demonstrated significantly higher 5-year OS and 3-year RFS rates (47.2% and 47.5%) compared to patients without HIPEC (34.5% and 21.3%). Stratification according to Peritoneal Surface Disease Severity Score (PSDSS) showed improved OS and RFS for patients with lower PSDSS (I–II) in the HIPEC-treated group ($p = 0.033$ and $p = 0.042$, respectively). The Clavien–Dindo classification of adverse event grades revealed no significant differences between HIPEC and controls ($p = 0.482$). While overall results were not statistically significant, our long-term follow-up emphasized the potential benefit of HIPEC-associated cytoreduction with paclitaxel, particularly in selected ovarian cancer patients with lower PSDSS indices.

**Keywords:** advanced or recurrent ovarian cancer; HIPEC; paclitaxel; cytoreduction; peritoneal metastases; peritoneal surface disease severity score

## 1. Introduction

Epithelial ovarian cancer (EOC) is one of the most fatal gynecological malignancies, and is particularly prevalent among elderly individuals, with a higher incidence in the sixth and seventh decades of life. Most patients with EOC present with peritoneal metastasis at the time of diagnosis [1].

Standard treatment for advanced-stage ovarian cancer involves cytoreductive surgery (CRS) in combination with platinum-based chemotherapy. Optional locoregional therapies within the abdominal cavity may also be included [2].

The peritoneal spread and distribution of ovarian cancer make this malignancy an ideal target for intraperitoneal chemotherapy (IP). IP capitalizes on the pharmacokinetic and pharmacodynamic advantages of the drugs administered, allowing high intraperitoneal drug concentrations with minimal systemic absorption due to the role played by the peritoneal–plasma barrier [3]. This strategy enables a direct, localized cytotoxic effect on tumor cells [4].

The addition of hyperthermia to standard chemotherapy heightens the therapeutic benefits of chemotherapy drugs due to the synergistic effect of the two treatments and induces production of heat shock proteins in cancer cells [5].

Hyperthermic intraperitoneal chemotherapy (HIPEC) may be performed intraoperatively following CRS as part of a single surgical intervention. The HIPEC procedure can have a duration of 30, 60, 90, or 120 min and be performed by the open (coliseum) or closed abdomen technique [6], although many different approaches for HIPEC have been described in the literature [7].

The addition of HIPEC after complete cytoreduction has been shown to be effective in treating advanced and recurrent EOC [8–10].

Paclitaxel, which has optimal pharmacokinetic properties for intraperitoneal administration, is sustained at high concentrations within the peritoneal cavity compared to plasma levels [11]. This property is partly attributable to its high molecular weight (853.906 g/mol), which impedes penetration through the peritoneal barrier, thereby resulting in minimal systemic absorption [8].

While the synergy between paclitaxel and hyperthermia remains a topic of debate, the combined use of paclitaxel and hyperthermia in HIPEC therapy has demonstrated both safety [12] and efficacy [13,14] in the treatment of patients with peritoneal metastases of ovarian origin. Although research has suggested using paclitaxel as a single drug for HIPEC in these patients [13–16], to date, no randomized clinical trials have evaluated this approach.

This clinical trial aims to evaluate the efficacy and safety of HIPEC with paclitaxel after complete cytoreduction in EOC with peritoneal involvement, regardless of primary or recurrent origin. Focus is placed on assessing the potential superiority of HIPEC using paclitaxel compared to standard treatment. HIPEC was performed in accordance with the closed abdomen technique and using the $CO_2$ recirculation system developed by our group in 2014, which enhanced temperature homogeneity and optimized solution distribution [17–19].

Patient selection, which is critical to the success of HIPEC therapy in treating peritoneal metastases, remains a primary challenge [20]. The Peritoneal Surface Disease Severity Score (PSDSS) is a predictive tool that identifies patients likely to benefit from aggressive treatments such as cytoreduction and HIPEC. This scoring system assesses clinical symptoms, the extent of peritoneal metastases (Peritoneal Carcinomatosis Index, PCI), and primary tumor histopathology to categorize patients into any of the four scores indicated by the PSDSS (Supplementary Table S1). Notably, this index has proven effective in defining peritoneal carcinomatosis of ovarian origin [21,22]. In the present study, we used the PSDSS to stratify patients within both groups and conducted analyses of the primary study objectives based on this index.

## 2. Materials and Methods

### 2.1. Patients and Trial Design

HIPECOVA is a single-center, randomized phase 3 clinical trial conducted in women with peritoneal involvement of primary EOC (International Federation of Gynecology and Obstetrics (FIGO) stages II, III, and IV) [23] or tumor recurrence, in whom complete cy-

toreduction was achieved. Randomization was performed intraoperatively once complete cytoreduction was successful.

Patients who were ineligible because of the extent of their peritoneal disease (assessed by diagnostic imaging) and where complete cytoreduction was not possible received neoadjuvant chemotherapy with paclitaxel (175 mg/m$^2$) and carboplatin (6 AUC). Subsequently, patients in whom a response to therapy was evidenced on imaging studies obtained after 3 cycles were eligible for interval surgery. Since the purpose of this research was to assess the added benefit of HIPEC with paclitaxel, patients with complete cytoreduction observed during interval surgery were also included and randomized intraoperatively.

This clinical trial was designed and performed by the multidisciplinary team delivering treatment for ovarian cancer and peritoneal carcinomatosis at General University Hospital of Ciudad Real. The study protocol was approved by the local ethics committee (no. 2011-006319-69, date of approval 25 April 2012). The HIPECOVA trial was authorized by the Spanish Drug Agency (EudraCT 2011006319-69) and registered in the ClinicalTrials.gov database (NCT02681432).

The trial's inclusion and exclusion criteria described below are available on the ClinicalTrials.gov registry website (https://www.clinicaltrials.gov/study/NCT02681432) (accessed on 15 December 2023).

### 2.2. Inclusion Criteria

- Female patients were eligible for inclusion if they were between 18 and 80 years of age and presented with histologically confirmed primary or recurrent EOC with peritoneal involvement. Women of childbearing age were required to have a negative pregnancy test to take part.
- Completeness of cytoreduction score (CC): CC0 (no visible residual tumor after surgery) or CC1 (less than 0.25 cm).
- No extra-abdominal tumor disease.
- Absence of heart failure. Adequate renal and hepatic function.
- Eastern Cooperative Oncology Group performance status of 0–2 or Karnofsky score ≥70%.

The completeness of the cytoreduction surgery was a determining factor in the patient inclusion criteria. Complete cytoreduction was defined by Jónsdóttir et al. as CC0 (no residual disease) and CC1 (residual tumor < 2.5 mm) [24].

### 2.3. Exclusion Criteria

- Patients with unresectable disease or incomplete cytoreduction.
- Contraindications for treatment with paclitaxel: patients with severe hypersensitivity to paclitaxel or any of its excipients, pregnancy or lactation, and patients with baseline neutrophil counts <1500/mm$^3$ (<1000/mm$^3$ for patients with Kaposi sarcoma). Paclitaxel is also contraindicated in patients with concurrent severe infections such as the following:
- Extra-abdominal metastases or unresectable liver metastases;
- Presence of other malignant tumor disease;
- Multi-segmental complete bowel obstruction;
- Patients with severe medical conditions precluding compliance with the study or that introduce an unacceptable risk;
- Patients who refuse treatment or do not consent to participate in the study.

### 2.4. Treatment Groups and Follow-Up

- HIPEC arm: CRS and hyperthermic intraperitoneal chemotherapy with paclitaxel (175 mg/m$^2$) for 60 min at a temperature of 42–43 °C (closed abdomen technique) followed by postoperative systemic intravenous (IV) chemotherapy with carboplatin (AUC = 6) and paclitaxel (175 mg/m$^2$) for 6 cycles.

- Non-HIPEC arm: CRS followed by postoperative systemic IV chemotherapy with carboplatin (AUC = 6) and paclitaxel (175 mg/m$^2$) for 6 cycles.

HIPEC was performed using a closed abdomen and $CO_2$ recirculation system. When required, anastomosis was completed before the HIPEC procedure. Following CRS, inflow and outflow catheters were positioned in the upper and lower abdominal cavities, respectively. Additionally, a $CO_2$ inflow catheter was placed in the right abdominal cavity. After inserting these catheters and a gas exchange device into the abdominal wall, the skin was closed to enable application of the paclitaxel solution, covering the entire peritoneal surface (Supplementary Figure S1).

The perfusion solution consisted of 1.36% glucose and 25 mmol/L bicarbonate (peritoneal dialysis solution). During the HIPEC procedure, a perfusion of chemotherapy and $CO_2$ was recirculated into the abdominal cavity, generating a turbulent flow to enhance the drug distribution across both visceral and peritoneal surfaces. Following the treatment, the abdominal cavity was opened, washed, and closed. Subsequently, all patients were closely monitored in the Intensive Care Unit (ICU) for the initial 2–3 days.

After hospital discharge, patients underwent regular follow-up, including physical examinations, CA-125 measurements, and computed tomography or positron emission tomography scans every 4 months for the first 2 years, followed by assessments every 6 months up to 3 years.

In cases of suspected recurrent disease, the diagnosis was based on the findings of imaging tests and elevation of the tumor marker CA-125. The existence of recurrence was subsequently confirmed by histological examination.

### 2.5. Endpoints

Primary endpoints included recurrence-free survival (RFS) and overall survival (OS). RFS was defined as the time from randomization to disease recurrence, defined according to the Response Evaluation Criteria in Solid Tumors. OS was defined as the time elapsed between randomization and the date of death or the end of the study.

Secondary endpoints included postoperative complications (adverse events within 30 days post-operatively) defined according to the National Cancer Institute criteria and the Common Terminology Criteria for AE (CTCAE).

### 2.6. Statistical Analysis

The study sample was calculated considering a 0.05 alpha risk, 0.20 beta risk, and difference of survival rate of 0.3 between arms based on the recurrence-free survival and using the method of sample size calculation for a log-rank test, indicating a requirement of 47 patients per arm to detect statistically significant differences ($p < 0.05$). Throughout the study period, 102 patients were assessed for eligibility, although the target sample size could not be reached due to a lower-than-expected participation rate. Consequently, the randomized sample comprised 76 patients, which carries with it a potential loss of significance. Subsequent exclusions reduced the final analysis to 32 patients in the experimental group and 23 in the control group.

Data collection was carried out by a group of data managers belonging to the research team and using an electronic database.

Statistical analysis was performed using the SPSS software package for Windows (IBM SPSS Statistics v.24) (International Business Machines Corporation, Armonk, NY, USA). Descriptive analyses were undertaken using means and standard deviations for quantitative variables and absolute and relative frequencies for qualitative variables. The Pearson Chi-square test was used to analyze the association between qualitative variables. Finally, the survival analysis was performed using the Kaplan–Meier method and Cox regression analysis, and the log-rank test was used to study the statistical differences between survival curves in qualitative variables. The median length of follow-up was estimated using the reverse Kaplan–Meier method.

## 3. Results

### 3.1. Patient Enrollment and Surgery-Related Characteristics

This single-center trial involved 102 eligible patients identified from August 2012 to December 2019. Of these, 90 provided preoperative consent, with 76 patients randomized intraoperatively into the control (*n* = 35) and experimental groups (*n* = 41). Further exclusions reduced the final analysis to 32 patients in the experimental group and 23 in the control group. Figure 1 contains a detailed CONSORT flow diagram depicting patient exclusions and allocations.

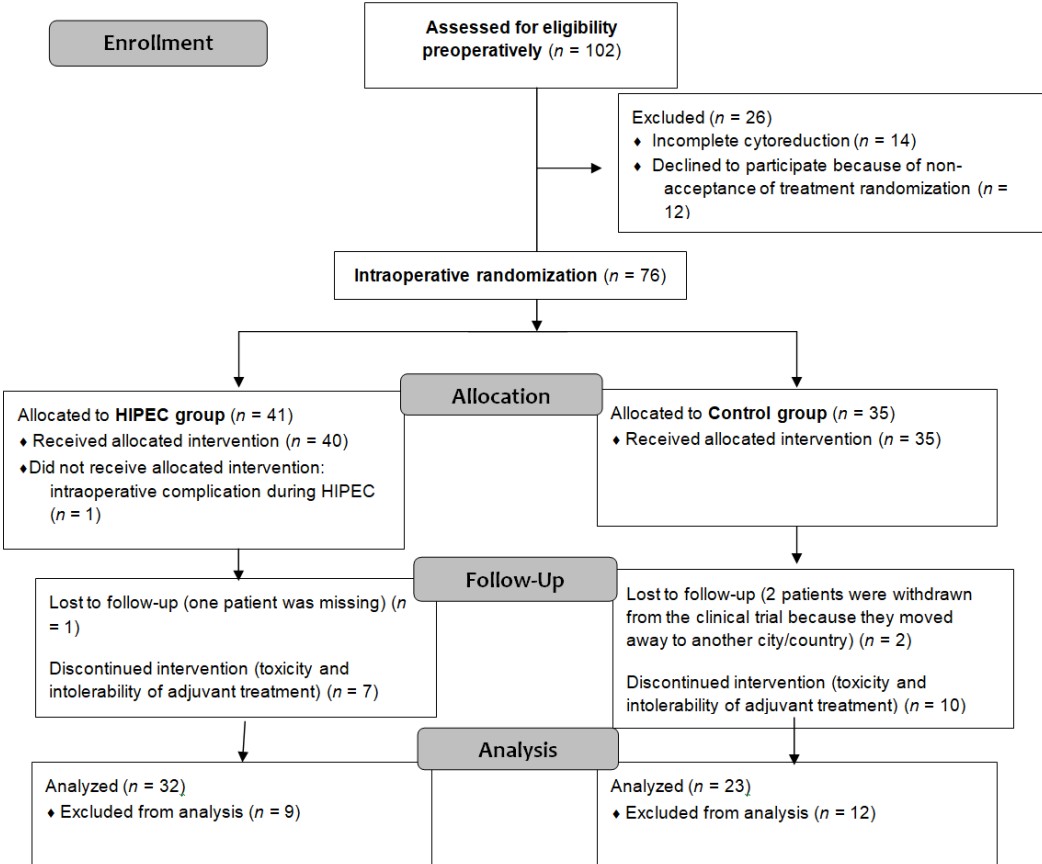

**Figure 1.** CONSORT flow diagram.

Baseline demographics showed no significant differences between the groups, ensuring homogeneity in the baseline variables (Table 1). FIGO stage IIIC was the most common stage observed (54.6%). Tumors with high-grade serous histology were the most frequently observed (63.6%).

No significant differences in PCI values were noted between the two arms, either before or during CRS (*p* = 0.666 and *p* = 0.720, respectively). The PCI before surgery and during CRS demonstrated associations with RFS (hazard ratio (HR) = 1.074 (95% CI, 1.003 to 1.150; *p* = 0.04)) and HR = 1.069 (95% CI, 1.015 to 1.125; *p* = 0.011), respectively.

Surgical procedures included upfront CRS in 26 patients (14 in the no HIPEC arm, 12 in the HIPEC arm), interval CRS in 26 patients (19 in the no HIPEC group, 7 in the HIPEC group), and secondary surgery in 3 patients (2 in the no HIPEC arm, 1 in the HIPEC arm).

The duration of surgery was significantly longer in the HIPEC arm (352.3 ± 98.3 min) compared to the no HIPEC arm (280 ± 73.48 min) due to the additional 60 minutes required for the HIPEC procedure.

An association was found between the number of peritonectomy procedures and OS (HR = 1.150; 95% CI, 1.039 to 1.274; *p* = 0.007) and with RFS (HR = 1.172; 95% interval CI, 1.035 to 1.327; *p* = 0.012).

In terms of the cytoreduction score reached in the included patients, 52 patients achieved complete cytoreduction (CCR-0), while 3 patients, with PCI value over 20, had a residual tumor of less than 2.5 mm (CCR-1).

**Table 1.** Baseline characteristics of patients.

| | **HIPEC** | **NO HIPEC** | ***p*-Value** |
|---|---|---|---|
| AGE (years) | 60.34 (±11.7) | 60.22 (±12.93) | 0.969 |
| BMI (kg/m$^2$) | 27.96 (±4.63) | 26.46 (±4.62) | 0.255 |
| AH | 15 (27.3%) | 6 (10.9%) | 0.118 |
| DM | 6 (10.9%) | 2 (3.6%) | 0.446 |
| ASA<br>- I<br>- II<br>- III<br>- IV | - 3 (5.5%)<br>- 18 (32.7%)<br>- 11 (20%)<br>- 0 (0%) | - 4 (7.3%)<br>- 14 (25.5%)<br>- 4 (7.3%)<br>- 1 (1.8%) | 0.3 |
| TUMOR HISTOLOGIC TYPE<br>- Serous<br>- Mucinous<br>- Endometrioid<br>- Clear-cell<br>- Other | <br>- 24 (43.6%)<br>- 3 (5.5%)<br>- 2 (3.6%)<br>- 2 (3.6%)<br>- 1 (1.8%) | <br>- 16 (29.1%)<br>- 1 (1.8%)<br>- 2 (3.6%)<br>- 3 (5.5%)<br>- 1 (0.8%) | 0.878 |
| SURGERY TYPE<br>- Primary surgery<br>- Interval surgery<br>- Secondary surgery | <br>- 12 (21.8%)<br>- 19 (34.5%)<br>- 1 (1.8%) | <br>- 14 (25.5%)<br>- 7 (12.7%)<br>- 2 (3.6%) | 0.118 |
| HISTOLOGIC GRADE<br>- Low grade<br>- Intermediate grade<br>- High grade | <br>- 4 (7.3%)<br>- 9 (16.4%)<br>- 19 (34.5%) | <br>- 0 (0%)<br>- 7 (12.7%)<br>- 16 (29.1%) | 0.234 |
| LYMPH NODES<br>- Positive<br>- Negative | <br>- 13 (23.6%)<br>- 19 (34.5%) | <br>- 11 (20%)<br>- 12 (21.8%) | 0.783 |
| FIGO STAGE<br>- IIa<br>- IIb<br>- IIIa<br>- IIIb<br>- IIIc<br>- IVa<br>- IVb | <br>- 3 (5.5%)<br>- 2 (3.6%)<br>- 1 (1.8%)<br>- 1 (1.8%)<br>- 16 (29.1%)<br>- 3 (5.5%)<br>- 6 (19.9%) | <br>- 2 (3.6%)<br>- 1 (1.8%)<br>- 1 (1.8%)<br>- 2 (3.6%)<br>- 14 (25.5%)<br>- 1 (1.8%)<br>- 2 (3.6%) | 0.929 |
| CT-PET PCI<br>- <10<br>- 11–20<br>- >20 | <br>- 19 (35.2%)<br>- 11 (20.4%)<br>- 1 (1.9%) | <br>- 14 (25.9%)<br>- 7 (13%)<br>- 2 (3.7%) | 0.666 |
| Surgery PCI<br>- <10<br>- 11–20<br>- >20 | <br>- 22 (40%)<br>- 8 (14.5%)<br>- 2 (3.6%) | <br>- 14 (25.9%)<br>- (14.5%)<br>- 1 (1.8%) | 0.72 |

BMI: Body Mass Index; AH: Arterial Hypertension; DM: Diabetes Mellitus; CT: Computerized Tomography; PCI: Peritoneal Cancer Index.

### 3.2. Survival Outcomes

The median length of follow-up was 32 months. Throughout this period, 50.9% of the patients experienced disease recurrence, while 34.5% died due to tumor progression. The

survival analysis results for key surgical outcome variables are presented in Supplementary Table S2.

The median OS (48 vs. 46 months, $p = 0.579$) and RFS (23 vs. 19 months, $p = 0.22$) showed no significant differences between the HIPEC and no HIPEC groups, respectively (Figure 2). However, the HIPEC group exhibited notably higher 5-year OS and 3-year RFS rates (47.2% and 47.5%, respectively) compared to the no HIPEC group (34.5% and 21.3%, respectively).

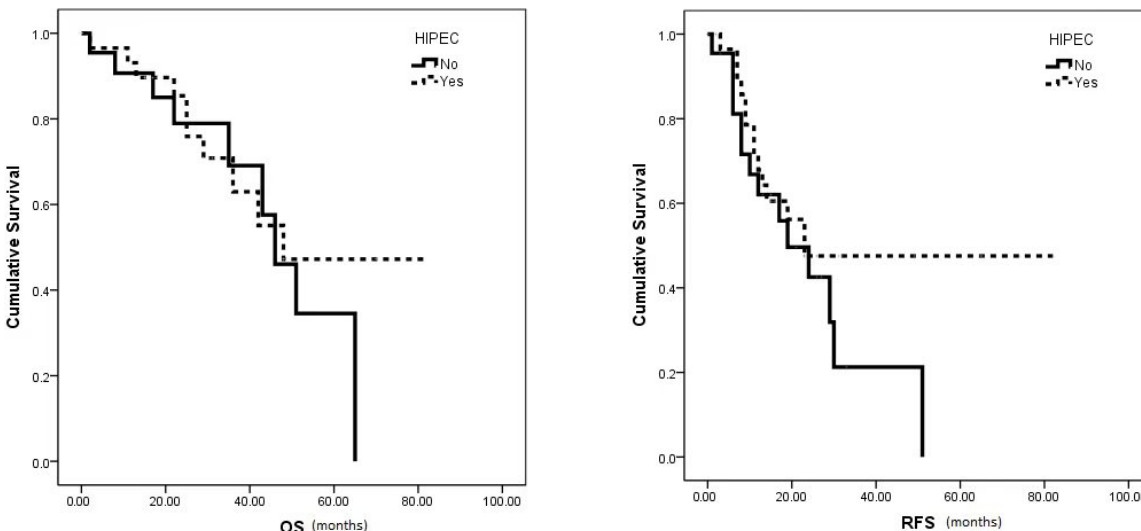

**Figure 2.** Median OS and RFS survival in HIPEC vs. no HIPEC group.

Treatment with HIPEC did not increase overall survival in patients treated with primary cytoreduction ($p = 0.246$) or interval surgery ($p = 0.584$) versus the control group.

HIPEC treatment also did not improve RFS in patients treated with primary cytoreduction ($p = 0.234$), interval surgery ($p = 0.242$), or secondary cytoreduction ($p = 0.157$) versus the control group.

### 3.3. Subgroup Analysis Based on PSDSS Score

Within the HIPEC group, patients with a PSDSS of I or II demonstrated notably higher OS and RFS compared to patients with scores of III or IV ($p = 0.033$ and $p = 0.042$, respectively) (Figure 3). Conversely, in the no HIPEC group, no differences were observed in RFS between patients with PSDSS I–II versus III–IV ($p = 0.310$).

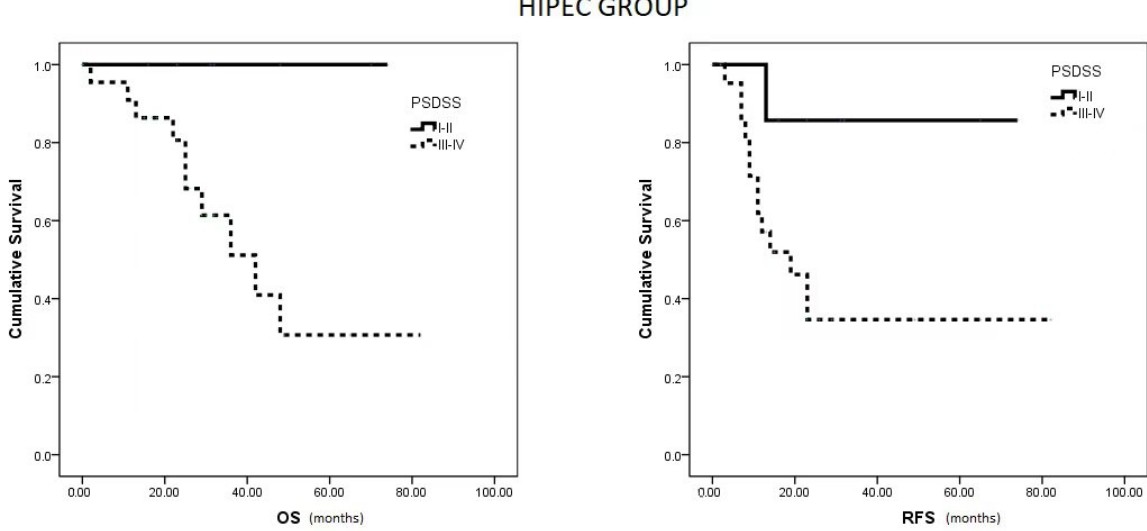

**Figure 3.** Median OS and RFS in HIPEC group based on PSDSS Score.

*3.4. Adverse Events*

According to the Clavien–Dindo classification, minor complications (grades I–IIIa) were observed in 26 patients (47.2%): 11 in the HIPEC group and 15 patients not undergoing HIPEC. Hematological complications were found to be the most prevalent adverse event, affecting 24.1% of patients (14.8% in the no HIPEC arm and 9.3% in the HIPEC arm, $p = 0.74$).

Severe complications (grades IIIb–V) were observed in 7 patients (12.7%), including 5 in the HIPEC group and 2 in the no HIPEC group. Among these, 3 patients (5.6%) experienced grade V adverse events involving multi-organ failure after postoperative perforation or intestinal leakage (1 in the no HIPEC arm and 2 in the HIPEC arm).

Additionally, 22 patients (40%) experienced no adverse events within 30 days post-surgery. Notably, there were no significant differences in adverse events of any grade observed between the two groups ($p = 0.482$) (Supplementary Table S3).

## 4. Discussion

Meta-analyses of published evidence consistently find that complete macroscopic removal of tumor disease is the main prognostic factor for improving both OS and RFS in advanced ovarian cancer [25].

In patients with recurrent ovarian cancer, surgical cytoreduction plays a pivotal role in OS [26].

In recent years, the integration of HIPEC therapy into primary cytoreduction, interval surgery, or secondary surgery for advanced ovarian cancer has gained increasing attention despite the lack of standardized protocols. Table 2 summarizes the outcomes of the major randomized clinical trials conducted to date.

**Table 2.** HIPEC randomized clinical trials in ovarian cancer.

| Author and Year | Clinical Trial | N | Surgery | HIPEC Drug ($mg/m^2$) | HIPEC (minutes) | Median PFS or DFS or RFS (months) | Median OS (months) | *p*-Value |
|---|---|---|---|---|---|---|---|---|
| Lim (2022) [27] | 2 center Phase III KOV-HIPEC-01 | 184 | Primary and Interval | Cisplatin 75 | 90 | 19.8 vs. 18.8 | 69.5 vs. 61.3 | $p > 0.05$ |
| | | | Interval subgroup | | | 17.4 vs. 15.4 | 61.8 vs. 48.2 | $p < 0.05$ |
| Cascales (2022) [28] | Single-center Phase III CARCINOHIPEC | 71 | Interval | Cisplatin 75 | 60 | 18 vs. 12 | 52 vs. 45 | $p > 0.05$ |
| | | | Subgroup with suprameso-colic disease | | | 24.1 vs. 9.4 | | $p < 0.05$ |
| Zivanovic (2021) [29] | Single-center Phase II | 98 | Secondary for platinum-sensitive | Carboplatin 800 | 90 | 15.7 vs. 12.3 | 59.7 vs. 52.5 | $p \geq 0.05$ |
| Driel (2018) [9] | Multicenter Phase III OVHIPEC | 245 | Interval | Cisplatin 100 | 90 | 14.2 vs. 10.7 | 45.7 vs. 33.9 | $p < 0.05$ |
| Spiliotis (2015) [10] | Single-center Phase III | 120 | Secondary for platinum-sensitive and resistant | Cisplatin 100 Paclitaxel 175 Doxorubicin 35 Mitomycin 15 | 60 | | 26.7 vs. 13.4 | $p < 0.05$ |

Abbreviations: N, number of patients; PFS, progression-free survival; DFS, disease-free survival; RFS, recurrence-free survival; OS, overall survival.

Notably, HIPEC therapy in advanced ovarian cancer has shown significant efficacy, particularly in association with interval surgery, cisplatin being commonly used as the primary HIPEC drug [9,27,28]. For the treatment of recurrent ovarian cancer, HIPEC therapy combined with secondary surgery has involved multiple drug regimens, leading to more varied and inconclusive outcomes [10,29].

The Dutch OVHIPEC clinical trial, a multicenter study of 245 patients, demonstrated that adding HIPEC (cisplatin 100 mg/m$^2$ at 40 °C over 90 min) to interval surgery after neoadjuvant treatment increased RFS (14.2 vs. 10.7 months, HR = 0.66, $p$ = 0.003) and OS (45.7 vs. 33.9 months, HR = 0.67, $p$ = 0.02) in primary ovarian cancer (FIGO III) [9].

A Korean randomized clinical trial (NCT010191636) including 184 patients with advanced ovarian cancer (FIGO III and IV) undergoing primary or interval surgery found no survival advantage in terms of OS (69.5 vs. 61.3 months, $p$ = 0.52) or progression-free survival (PFS) (19.8 vs. 18.8 months, $p$ = 0.43). However, in the subgroup undergoing interval surgery after neoadjuvant chemotherapy, the addition of HIPEC (cisplatin 75 mg/m$^2$ at 41.5 °C over 90 min) showed favorable outcomes, presenting improved PFS (17.4 vs. 15.4 months, $p$ = 0.04) and OS (61.8 vs. 48.2 months, $p$ = 0.04). Conversely, for patients undergoing primary CRS, HIPEC did not demonstrate the same benefits [27].

In a prospective phase 3 clinical trial in Spain led by Cascales et al., the efficacy of interval CRS combined with HIPEC using cisplatin (75 mg/m$^2$ for 60 min at 42 °C) was investigated in patients with advanced ovarian cancer and peritoneal metastases. Among the 71 participants, 36 underwent interval surgery, while 35 received interval surgery coupled with HIPEC. The primary endpoint was disease-free survival (DFS), with secondary endpoints encompassing OS, morbidity, and quality of life (QoL). No statistically significant differences were observed in median DFS, median OS, or overall morbidity rates between the groups. Nevertheless, in the subgroup of patients with disease presence in the supramesocolic compartment, administration of HIPEC was linked to improved DFS (9.4 months in the control group and 24.1 months in the experimental group; $p$ = 0.031). Interestingly, the incorporation of HIPEC had no notable impact on patient QoL across the evaluated dimensions [28].

The randomized phase II clinical trial by Zivanovic et al. involved 98 patients with recurrent advanced ovarian cancer. The trial used carboplatin at a dose of 800 mg/m$^2$ for 90 minutes as the HIPEC drug. While the median PFS was 15.7 months compared to 12.3 months, and the median OS was 59.7 versus 52.5 months, the observed differences were not statistically significant ($p$ = 0.05 and 0.32, respectively). Despite being well-tolerated, the use of HIPEC with carboplatin did not result in superior clinical outcomes [29].

In contrast, the Greek clinical trial, involving 120 patients and conducted as a single-center study, revealed that combining HIPEC (at 42.5 °C for 60 min) with CRS for recurrent ovarian cancer led to a significant increase in median survival (26.7 months vs. 13.4 months, $p$ < 0.006) and 3-year survival rates (75% vs. 18%). Various HIPEC drugs were employed, such as cisplatin (100 mg/m$^2$) and paclitaxel (175 mg/m$^2$) for platinum-sensitive disease, and doxorubicin (35 mg/m$^2$) and paclitaxel (175 mg/m$^2$) or mitomycin (15 mg/m²) for platinum-resistant disease [10].

While platinum-based compounds, particularly carboplatin and cisplatin, along with paclitaxel, are the standard chemotherapeutic agents for first-line treatment in ovarian cancer, cisplatin has historically been the drug of choice in HIPEC for advanced ovarian cancer. However, the favorable pharmacokinetic properties and promising outcomes of paclitaxel for treating peritoneal carcinomatosis originating from ovarian cancer [14] led us to conduct a clinical trial aimed at assessing its efficacy. The HIPECOVA trial was designed to investigate the impact of HIPEC with paclitaxel following surgical treatment for peritoneal metastases in both primary and recurrent advanced ovarian cancer. Our study primarily focused on evaluating the effects of the treatment on RFS and OS. Although the trial did not definitively establish the superiority of HIPEC therapy involving paclitaxel after cytoreduction in enhancing survival outcomes for advanced ovarian cancer, whether applied during primary, interval, or secondary surgeries for relapse, our findings emphasize the need for a thorough and comprehensive evaluation.

Based on available data, anticipated outcomes in the control group for advanced ovarian cancer indicate a median RFS range of 11 to 16.4 months and a median OS ranging between 23 and 44.3 months [29]. However, the results among patients in the HIPEC group were more favorable, achieving a median RFS and OS of 23 and 48 months, respectively, suggesting notably improved outcomes, which can be deemed optimal.

When considering the adverse events associated with CRS and HIPEC, this combined treatment consistently demonstrates varying perioperative mortality rates, spanning from 0% to 18%, along with morbidity rates that fall between 30% and 70% [30]. In our study, grades IIIa–IIIb adverse events were observed in 14.6% of patients, while no grade IV events were reported. Notably, no significant differences in adverse events of any grade were observed between the groups ($p = 0.482$), which is consistent with other clinical trials [9]. The observed complication rates remain comparable to those reported in other studies. Hematological complications were noted in 24.1% of cases, consistent with the reported incidence range of hematological toxicity, which spans from 4% to 39%. Additionally, 2.2% of patients developed intestinal leakage, which is lower than the described range for grade III/IV gastrointestinal complications, typically reported between 4.5% and 19% [30]. In summary, this randomized clinical trial did not reveal a significantly different incidence of adverse events compared to similar studies.

Strengths: One of the primary strengths of this clinical trial lies in its innovative approach, as it employs paclitaxel as a single-agent HIPEC, a relatively uncommon but highly promising treatment strategy for advanced or recurrent ovarian cancer. The pharmacokinetic profile of paclitaxel lends itself to intraperitoneal administration, potentially enhancing its therapeutic efficacy in this context. Moreover, the single-center design of this trial, together with the fact that the study was managed by a cohesive research team, enhances its reliability by minimizing potential biases and ensuring a uniform approach to data collection and analysis.

Limitations: Despite these strengths, the study has several limitations. Firstly, the number of patients enrolled was low, owing to multiple factors such as patient reluctance to accept randomized treatments, disparities between intraoperative findings and definitive histopathological results, and non-adherence to established monitoring protocols. This limited sample size may affect the generalizability of the results and the statistical power of the study. Additionally, the assumption that HIPEC therapy was considered to be effective solely based on surgical control of peritoneal disease, irrespective of its association with primary, interval, or secondary surgeries, may have led to an overestimation of treatment effectiveness. Implementing randomized stratification may have improved the quality of this research and yielded more robust outcomes.

## 5. Conclusions

The outcomes of this initial phase III randomized clinical trial evaluating the efficacy of HIPEC with paclitaxel following cytoreduction in advanced ovarian cancer reveal no statistically significant differences between the treatment and control groups. Despite the overall results lacking statistical significance, our extended follow-up reveals a sustained positive trend in both overall survival and disease-free survival. Notably, this benefit is particularly evident in selected ovarian cancer patients with lower Peritoneal Surface Disease Severity (PSDSS) indices, suggesting a potential benefit of cytoreduction associated with HIPEC with paclitaxel. This promising observation warrants further investigation to better understand and confirm its clinical implications.

**Supplementary Materials:** The following supporting information can be downloaded at https://www.mdpi.com/article/10.3390/curroncol31020048/s1: Figure S1: Closed abdomen and $CO_2$ recirculation system; Table S1: Peritoneal Surface Disease Severity Score (PSDSS); Table S2: Results of the survival analysis of the main variables related to surgery outcomes; Table S3: Complications according to the Clavien–Dindo classification.

**Author Contributions:** Conceptualization, P.V.C. and D.P.-V.; Methodology, C.L.d.l.M., D.P.-V. and F.J.R.C.; Software, A.A.; Validation, F.J.R.C. and J.M.; Formal analysis, M.A.-S.; Investigation, S.S.G., A.A. and E.G.S.; Resources, C.L.d.l.M. and J.M.; Data curation, S.S.G., A.A., D.P.-V. and E.G.S.; Writing—original draft preparation, P.V.C., M.A.-S. and S.S.G.; Writing—review and editing, P.V.C., S.S.G., E.G.S., D.P.-V., J.M., A.A., C.L.d.l.M., F.J.R.C. and M.A.-S.; Visualization, D.P.-V. and F.J.R.C.; Supervision, J.M. and C.L.d.l.M.; Project management, S.S.G., C.L.d.l.M. and P.V.C.; Funding acquisition, P.V.C. All authors have read and agreed to the published version of the manuscript.

**Funding:** This study was funded by Instituto de Salud Carlos III (ISCIII) through the project PI20/01052. The General University Hospital of Ciudad Real has also contributed its own resources to conduct this clinical trial.

**Institutional Review Board Statement:** The present study was approved by the Research Ethics Committee of the University General Hospital of Ciudad Real (no. 2011-006319-69, date of approval 25 April 2012). The procedures followed were in accordance with the Code of Ethics of the World Medical Association (Declaration of Helsinki) and with the protocols and requirements established by the Spanish Agency of Medicines and Medical Devices.

**Informed Consent Statement:** Informed consent was obtained from all subjects involved in the study.

**Data Availability Statement:** The datasets analyzed during the current study are available from the corresponding author upon reasonable request.

**Acknowledgments:** The authors would like to thank Javier Sánchez Menor and Pilar Marta for their technical assistance and Oliver Shaw for revising the manuscript in aspects related to the English language. We are also grateful to Jesús Fernández Sanz and Alberto Martínez Albalat for their support of this project.

**Conflicts of Interest:** The authors declare no conflicts of interest. The funders had no role in the design of the study; in the collection, analyses, or interpretation of data; in the writing of the manuscript; or in the decision to publish the results.

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
