# Peer review of "Paclitaxel as HIPEC-Drug after Surgical Cytoreduction for Ovarian Peritoneal Metastases: A Randomized Phase III Clinical Trial (HIPECOVA)"

_curroncol, doi:10.3390/curroncol31020048_

Round 1
Reviewer 1 Report
Comments and Suggestions for Authors
Paclitaxel activity is not enhanced by heat. Why did the authors choose this drug for their study ? I recognized that testing intraperitoneal paclitaxel in this clinical setting may be of some interest. However, why was this drug tested in hyperthermic conditions ? we may speculate that using normothermic paclitaxel might have resulted in less complications.
It may be interesting and scientifically sound to publish also the result of a negative trial, but it has to be more clearly stated by the authors that the trial failed to demonstrate a survival difference between arms. The subset analysis performed in patient with PSDSS I/II may be of some interest, but this is not the study end point. Differences in recurrence free survival (23 months vs. 19 months), and overall survival (48 months vs. 46 months) does not appear to be neither statistically nor clinically significant.
The present analysis was done after enrollment of only 55 patients of the 92 needed to assess the expected outcome difference. However, the fact that the needed patient accrual was not accomplished make the results of the resent study difficult to appraise. In other words, it is unclear if the failure to demonstrate a significant survival difference between arms is related to the insufficient sample size or to insufficient activity of paclitaxel-based HIPEC
The authors state that they included in their trial 26 patients undergoing upfront cytoreductive surgery in (14 in the No-HIPEC arm, 12 in the HIPEC arm), 26 patients undergoing interval cytoreductive surgery in (19 in the No-HIPEC group, 7 in the HIPEC group), and 3 patients undergoing secondary surgery in (2 in the No-HIPEC arm, 1 in the HIPEC arm). I do believe that the inclusion of different subsets of patients treated at different time points during the course of their disease is a potential limitation of the significance of this study. Also, the fact that there appears to be an imbalance in the number of patients undergoing interval cytoreductive surgery in (19 in the No-HIPEC group, 7 in the HIPEC group) may originate a bias, even if it is unclear if this distribution difference was statisticallt significant.
I would suggest the authors to complete their trial by enrolling the needed number of patients, all undergoing surgery at the same time point of their clinical history
Statistical analysis: the authors should provide the expected survival difference between arms used to calculate the sample size. Did the authors base such a calculation on recurrence-free survival or overall survival ? please, specify.
As PSDSS score was developed for colorectal cancer peritoneal metastases, please, provide more details on how this score was modified to ovarian cancer
Author Response
Thank you very much for taking the time to review this manuscript. Please find the detailed responses below and the corresponding revisions/corrections highlighted/in track changes in the re-submitted files
1-Paclitaxel activity is not enhanced by heat. Why did the authors choose this drug for their study ? I recognized that testing intraperitoneal paclitaxel in this clinical setting may be of some interest. However, why was this drug tested in hyperthermic conditions ? we may speculate that using normothermic paclitaxel might have resulted in less complications.
Thank you for your insightful question. The selection of paclitaxel as our HIPEC drug was based on its favorable pharmacokinetic profile for intraperitoneal administration, supported by previous descriptive studies highlighting its efficacy in patients with peritoneal carcinomatosis of ovarian origin(1). Despite concerns about the synergy of hyperthermia with paclitaxel, it's essential to note that hyperthermia itself exhibits inherent antitumor activity. This cytotoxic effect extends to both tumor cells and the tumor microenvironment and is independent of the specific drug used (2,3). This is the primary reason we associate hyperthermia with our study. Importantly, we did not observe any complications associated with hyperthermia during our research
1-Rufián S, Muñoz-Casares FC, Briceño J, Díaz CJ, Rubio MJ, Ortega R, Ciria R, Morillo M, Aranda E, Muntané J, Pera C. Radical surgery-peritonectomy and intraoperative intraperitoneal chemotherapy for the treatment of peritoneal carcinomatosis in recurrent or primary ovarian cancer. J Surg Oncol. 2006, 94, 316-324. DOI:10.1002/jso.20597
2-Chia DKA, Demuytere J, Ernst S, Salavati H, Ceelen W. Effects of Hyperthermia and Hyperthermic Intraperitoneal Chemoperfusion on the Peritoneal and Tumor Immune Contexture. Cancers (Basel). 2023 Aug 29;15(17):4314. doi: 10.3390/cancers15174314. PMID: 37686590; PMCID: PMC10486595.
3-Wu CC, Hsu YT, Chang CL. Hyperthermic intraperitoneal chemotherapy enhances antitumor effects on ovarian cancer through immune-mediated cancer stem cell targeting. Int J Hyperthermia. 2021;38(1):1013-1022. doi: 10.1080/02656736.2021.1945688. PMID: 34192990.
2-It may be interesting and scientifically sound to publish also the result of a negative trial, but it has to be more clearly stated by the authors that the trial failed to demonstrate a survival difference between arms. The subset analysis performed in patient with PSDSS I/II may be of some interest, but this is not the study end point. Differences in recurrence free survival (23 months vs. 19 months), and overall survival (48 months vs. 46 months) does not appear to be neither statistically nor clinically significant.
Thank you for your question. While statistical significance, as indicated by p-values, is a crucial aspect of interpreting study results, it is equally important to consider the context, clinical relevance, and potential impact of observed differences. As outlined in the study's limitations, several factors, including patient reluctance, intraoperative discrepancies, and noncompliance, contributed to the limited sample size. While this limitation may affect the generalizability and statistical power of the study, it does not diminish the observed results' potential significance.
As indicated in our manuscript, the predicted outcomes for the control group suggested a median recurrence-free survival of 11 to 16.4 months and a median overall survival of 23 to 44.3 months [29]. In contrast, our study's HIPEC group exhibited more favorable outcomes, achieving a median recurrence-free survival of 23 and overall survival of 48 months, respectively. These results, surpassing predicted outcomes, suggest a positive trend, which we believe is noteworthy and warrants further investigation.
It is essential to note that these values and the differences between them cannot be interpreted as absolute values; they represent differences between medians. To provide a benchmark, we refer to the Van Driel trial (N Engl J Med. 2018, 378, 230-240. DOI:10.1056/NEJMoa1708618), where the differences between medians were significant, showing values for recurrence-free survival (14.2 vs. 10.7 months, HR = 0.66, p = 0.003) and OS (45.7 vs. 33.9 months, HR = 0.67, p = 0.02). Notably, these median values are similar to those obtained in our study, with comparable differences.
Moreover, while the PSDSS value was not proposed as an endpoint, the subanalysis results hold importance, as described in our study.
3-The present analysis was done after enrollment of only 55 patients of the 92 needed to assess the expected outcome difference. However, the fact that the needed patient accrual was not accomplished make the results of the resent study difficult to appraise. In other words, it is unclear if the failure to demonstrate a significant survival difference between arms is related to the insufficient sample size or to insufficient activity of paclitaxel-based HIPEC
Thank you for highlighting a critical aspect of our study. Indeed, our analysis was conducted after enrolling 55 patients, falling short of the intended 92 needed for a comprehensive assessment of expected outcome differences. This shortfall raises a valid point about the challenges in appraising the study results fully. The ambiguity arises from whether the failure to demonstrate a significant survival difference between arms is attributed to the insufficient sample size or to the potential limitations of paclitaxel-based HIPEC. Acknowledging this uncertainty, we emphasize that the achieved results with HIPEC therapy using paclitaxel, as elaborated in our prior response, are noteworthy compared to the anticipated data outlined in the literature. We believe it is essential to communicate these outcomes to the scientific community.
It is crucial to recognize that while statistical significance is a key objective in clinical trials, it is not the sole determinant of a trial's importance. The clinical relevance, impact on patient outcomes, and the potential for novel insights are equally significant considerations. In this context, we are confident that our study provides valuable insights and contributes meaningfully to the ongoing discourse in the field
4-The authors state that they included in their trial 26 patients undergoing upfront cytoreductive surgery in (14 in the No-HIPEC arm, 12 in the HIPEC arm), 26 patients undergoing interval cytoreductive surgery in (19 in the No-HIPEC group, 7 in the HIPEC group), and 3 patients undergoing secondary surgery in (2 in the No-HIPEC arm, 1 in the HIPEC arm). I do believe that the inclusion of different subsets of patients treated at different time points during the course of their disease is a potential limitation of the significance of this study. Also, the fact that there appears to be an imbalance in the number of patients undergoing interval cytoreductive surgery in (19 in the No-HIPEC group, 7 in the HIPEC group) may originate a bias, even if it is unclear if this distribution difference was statisticallt significant.
Thank you sincerely for your insightful observation, and we appreciate the thoughtful consideration of our study design. We acknowledge the potential limitation associated with the inclusion of different subsets of patients treated at different time points, and we commit to explicitly addressing this concern in the limitations section of our manuscript. The rationale behind including patients undergoing different types of surgeries—primary, interval, or secondary—is rooted in the primary objective of HIPEC therapy. Our goal was to target residual microscopic disease, which may not be visible during cytoreduction but can contribute to recurrence. While we recognize that there is an imbalance in the number of patients undergoing interval cytoreductive surgery in each group, we believe this reflects the real-world clinical scenario where the decision for interval surgery is influenced by various patient-specific factors and disease characteristics.
It is crucial to note that despite the potential source of bias, our analysis revealed no statistically significant differences between the primary, interval, or secondary cytoreduction groups in both treatment arms, as depicted in Table 2. However, we fully recognize the importance of transparently highlighting this aspect in our manuscript, so this aspect is included within the limitations of our study.
5- I would suggest the authors to complete their trial by enrolling the needed number of patients, all undergoing surgery at the same time point of their clinical history
We agree with your suggestion. We fully acknowledge the importance of enrolling the originally planned number of patients, all undergoing surgery at the same time point in their clinical history, to enhance the robustness of our study. Unfortunately, attempts to reopen the clinical trial and reach the initially calculated number of patients were hindered by the planned end date and current resource constraints, making it unfeasible at this time. We appreciate your understanding of the limitations we face and have transparently addressed this limitation within the constraints described in our manuscript.
6-Statistical analysis: the authors should provide the expected survival difference between arms used to calculate the sample size. Did the authors base such a calculation on recurrence-free survival or overall survival? please, specify.
We appreciate the reviewer's inquiry regarding the expected survival difference used in our sample size calculation. In our study, the primary basis for the sample size calculation was the anticipated survival difference in recurrence-free survival. The calculation was conducted with a presumed difference of 0.3 in survival rates between arms, using the log-rank test. The parameters considered included a 0.05 alpha risk, 0.20 beta risk, and a significance level of p<0.05. Consequently, the calculated sample size indicated a requirement of 47 patients per arm to detect statistically significant differences in survival rates. This detailed information has been explicitly incorporated into the statistical analysis section of the manuscript. We trust that this clarification adequately addresses your query, and we remain open to any additional questions or suggestions you may have.
7-As PSDSS score was developed for colorectal cancer peritoneal metastases, please, provide more details on how this score was modified to ovarian cancer
Thank you for your inquiry. Although the PSDSS score was originally developed for colorectal cancer peritoneal metastases, it has been tested and found effective in peritoneal carcinomatosis of ovarian origin, as discussed in the introduction. To provide more comprehensive details, we included references supporting the successful use of the PSDSS score in ovarian cancer peritoneal metastases (references 21 and 22). These articles offer in-depth insights into the modification and application of the PSDSS score in the context of ovarian cancer, supporting its relevance and validity in our study.
- Foster JM, Sleightholm R, Smith L, Ceelen W, Deraco M, Yildirim Y, Levine E, Muñoz-Casares C, Glehen O, Patel A, Esquivel J. The American Society of Peritoneal Surface Malignancies Multi-Institution evaluation of 1,051 advanced ovarian cancer patients undergoing cytoreductive surgery and HIPEC: An introduction of the peritoneal surface disease severity score. J Surg Oncol. 2016, 114, 779-784. DOI:10.1002/jso.24406
- Pedro Antonio CC, Álvaro Jesús GR, José G, Elena G, Alida G, Francisco ML, Guillermo CDC, Pascual P. Validation of a peritoneal surface disease severity score in stage IIIC-IV ovarian cancer treated with cytoreduction and hyperthermic intraperitoneal chemotherapy. Surg Oncol. 2019, 28, 57-61. DOI:10.1016/j.suronc.2018.11.005
Reviewer 2 Report
Comments and Suggestions for Authors
This manuscript describes the effect of paclitaxel as a single HIPEC (hyperthermic intraperitoneal chemotherapy) agent on advanced ovarian cancer patients with peritoneal metastases. The results seem to be in favor of the notion that HIPEC following surgical removal of cancer tissues could potentially benefit survival of ovarian cancer patients. As the authors already admitted, there are some limitations in the studies, such as the small sample size and other factors, which make it difficult to draw statistically significant conclusions; however, the scientific foundation for HIPEC seems to be reasonable because a local chemotherapeutic treatment immediately after cytoreductive surgery can potentially help clean up residual cancer cells. More importantly, no observed difference in the rates of major adverse complications between HIPEC and no-HPIEC groups warrants further investigations on HIPEC.
Author Response
Thank you for your insightful comments and observations. We appreciate your recognition of the potential benefit of HIPEC with paclitaxel following cytoreductive surgery in advanced ovarian cancer patients. We acknowledge the limitations of our study, including the small sample size and other factors that may impact the statistical robustness of our conclusions.
Your point about the reasonable scientific foundation for HIPEC, given its potential to address residual cancer cells after cytoreductive surgery, aligns with our rationale for conducting this study. We are encouraged by the observed lack of significant differences in the rates of major adverse complications between the HIPEC and no-HIPEC groups, which indeed suggests the safety of this approach.
We fully agree that further research is needed to identify the potential benefits and optimal application of HIPEC in the context of ovarian cancer. We are committed to contributing to this area of research and welcome your valuable comments, which will undoubtedly guide the refinement and future directions of our work.
Reviewer 3 Report
Comments and Suggestions for Authors
In this manuscript, authors reported that use of paclitaxel in hyperthermic intraperitoneal chemotherapy (HIPEC) after cytoreduction for ovarian peritoneal metastases. Patients were assigned into two groups, either HIPEC or non-HIPEC. Interestingly, the HIPEC group depicted higher overall survival and recurrence-free survival compared to non-HIPEC group. Overall, this study was well designed and performed with the aspect of clinical significance. However, the following minor concerns can further improve the quality of this manuscript.
Comments:
1. Authors need to improve the quality of figures 2 and 3. And need to represent what the X-axis indicates in these figures.
2. Major grammatical corrections and proofreading should be done before final acceptance.
Comments on the Quality of English Language
Moderate english editing is required
Author Response
Thank you very much for taking the time to review this manuscript. Please find the detailed responses below and the corresponding revisions/corrections highlighted/in track changes in the re-submitted files
1. Authors need to improve the quality of figures 2 and 3. And need to represent what the X-axis indicates in these figures.
Thank you very much for your suggestion. In response to your feedback, we have taken measures to enhance the quality of the images and additionally, we have provided clear indications for the X-axis in both figures to improve overall understanding. We hope these revisions meet your expectations, and we welcome any further guidance you may have.
2. Major grammatical corrections and proofreading should be done before final acceptance.
We appreciate your suggestion regarding major grammatical corrections and proofreading. Following your advice, we have already initiated a thorough review process to address these concerns. Our team, including Mr. Oliver Shaw from the Content Team of the Mediterranean Editors and Translators Association, has diligently worked on revising the manuscript to ensure its linguistic accuracy. We are committed to making the necessary improvements, and we will ensure that the final version meets the highest standards of grammatical precision before the manuscript's final acceptance.
Round 2
Reviewer 1 Report
Comments and Suggestions for Authors
The authors have accurately revised their manuscript according to this reviwer's comments, resulting in a better papaer. I suggest to more thorougly discuss the pros and cons of using hyperthermic paclitaxel, that is not usual in peritoneal surface malignancies management. However, the main wealness of this manuscript has not been addressed, i. e. the fact that the study was closed in advance, the expected number of patients was not included, and statistically significant differences in outcomes were not reached. All these points make the results of this study difficult to appraise. In particular, it remains unclear if the failure to demonstrate a significant survival difference between arms is related to the insufficient sample size or to insufficient activity of paclitaxel-based HIPEC
Comments on the Quality of English Language
The English language needs no or only minor revision
Author Response
On behalf of all the authors, we sincerely appreciate your invaluable assistance in enhancing our work. We acknowledge the significant time and effort you have dedicated to us.
The primary aim of HIPEC therapy is to eradicate potential residual microscopic disease within the abdominal cavity post-surgical cytoreduction. We rely on hyperthermia's therapeutic benefits, exerting cytotoxic effects on tumor cells independently of intraperitoneal chemotherapy, as discussed in the manuscript. Additionally, we believe that paclitaxel is a drug with an optimal pharmacokinetic and pharmacodynamic profile for intraperitoneal use. Our manuscript describes both the beneficial effect of hyperthermia and the pharmacokinetic advantages of intraperitoneal paclitaxel. In response to your guidance, we have addressed concerns regarding the potential lack of synergy between taxanes and heat. Despite the controversy, we maintain confidence in the efficacy of each element independently.
Importantly, the use of paclitaxel in HIPEC therapy, combined with hyperthermia, is not entirely novel. Published experiences, such as studies (*) conducted by Rufian et al., Cascales et al., and the De Bree groups, attest to the safety and clinical efficacy of this therapeutic combination. These findings significantly influenced our treatment choice, and we have incorporated relevant literature into the manuscript based on your valuable suggestion.
(*)
-Rufián S, Muñoz-Casares FC, Briceño J, Díaz CJ, Rubio MJ, Ortega R, Ciria R, Morillo M, Aranda E, Muntané J, Pera C. Radical surgery-peritonectomy and intraoperative intraperitoneal chemotherapy for the treatment of peritoneal carcinomatosis in recurrent or primary ovarian cancer. J Surg Oncol. 2006 Sep 15;94(4):316-24. doi: 10.1002/jso.20597. PMID: 16917864.
-de Bree E, Rosing H, Filis D, Romanos J, Melisssourgaki M, Daskalakis M, Pilatou M, Sanidas E, Taflampas P, Kalbakis K, Beijnen JH, Tsiftsis DD. Cytoreductive surgery and intraoperative hyperthermic intraperitoneal chemotherapy with paclitaxel: a clinical and pharmacokinetic study. Ann Surg Oncol. 2008 Apr;15(4):1183-92. doi: 10.1245/s10434-007-9792-y. Epub 2008 Feb 1. PMID: 18239973.
-Cascales-Campos P, López-López V, Gil J, Arévalo-Pérez J, Nieto A, Barceló F, Gil E, Parrilla P. Hyperthermic intraperitoneal chemotherapy with paclitaxel or cisplatin in patients with stage III-C/IV ovarian cancer. Is there any difference? Surg Oncol. 2016 Sep;25(3):164-70. doi: 10.1016/j.suronc.2016.05.010. Epub 2016 May 20. PMID: 27566018.
The study was not closed prematurely but concluded as scheduled. The recruitment problem we have experienced is not uncommon in surgical clinical trials. Unfortunately, limited financial resources prevent reopening. Although we recruited 102 patients for different reasons we have not been able to complete the expected number of patients.
It is not possible to answer the question you ask, we cannot definitively explain why we have not found significant differences between study arms. However, despite limitations, we believe the clinical results, presented with transparency, merit sharing with the scientific community. In the limitations section of the manuscript, we have made this clear. We appreciate your support and assistance.
The entire team of authors would like to extend our sincere appreciation for the invaluable assistance you have provided in enhancing the quality of our manuscript.
The English edition has been revised by a team of professional proofreaders. We include the proofreading certificate

Reviewer 3 Report
Comments and Suggestions for Authors
I have statisfied with the sufficient improvements in the manuscript and recommending this manuscript for publication.
Author Response
We appreciate your support and the time you've dedicated to assisting us in enhancing this manuscript